# Towards Generative Latent Variable Models for Speech

## Abstract

While stochastic latent variable models (LVMs) now achieve state-of-the-art performance on natural image generation, they are still inferior to deterministic models on speech. On natural images, these models have been parameterised with very deep hierarchies of latent variables, but research shows that these model constructs are not directly applicable to sequence data. In this paper, we benchmark popular temporal LVMs against state-of-the-art deterministic models on speech. We report the likelihood, which is a much used metric in the image domain but rarely, and often incomparably, reported for speech models. This is prerequisite work needed for the research community to improve LVMs on speech. We adapt Clockwork VAE, a state-of-the-art temporal LVM for video generation, to the speech domain, similar to how WaveNet adapted PixelCNN from images to speech. Despite being autoregressive only in latent space, we find that the Clockwork VAE outperforms previous LVMs and reduces the gap to deterministic models by using a hierarchy of latent variables.

## 1 Introduction

With the introduction of the variational autoencoder (VAE) (Kingma & Welling, 2014; Rezende et al., 2014) came two rapid extensions for modeling speech data (Chung et al., 2015; Fraccaro et al., 2016). Since then, temporal LVMs have undergone little development and their autoregressive counterparts, such as the WaveNet (Oord et al., 2016a), remain state-of-the-art. In the image domain, generative LVMs have recently shown superior performance to the PixelCNN (Oord et al., 2016c;b; Salimans et al., 2017), the model that built the foundation for WaveNet. The improvements in performance have primarily been driven by altered inference models, including top-down (Sønderby et al., 2016) and bidirectional inference (Maaløe et al., 2019), deeper latent hierarchies and skip connections (Sønderby et al., 2016; Maaløe et al., 2019; Vahdat & Kautz, 2020; Child, 2021).

To innovate and compare LVMs we need good baselining, similar to the many reported benchmarks within the image domain. However, research in the speech domain has often omitted reporting a likelihood (Oord et al., 2016a; Hsu et al., 2017; Oord et al., 2018b) or has reported likelihoods that are incomparable due to subtle differences in the model parameterizations (Chung et al., 2015; Fraccaro et al., 2016; Hsu et al., 2017; Aksan & Hilliges, 2019). Without a proper comparison standard, it is difficult for the field of explicit likelihood models on speech to evolve further.

This research pushes forward the state of the LVM on speech by (i) properly benchmarking previous models, (ii) introducing a high-performing hierarchical temporal LVM architecture, and (iii) analyzing the representations of the latent variables. We find that:

(I) Previous state-of-the-art LVMs achieve close to identical likelihood performance, still significantly inferior to the WaveNet. Interestingly, we also find that the WaveNet performs almost identically to a standard LSTM parameterization (Hochreiter & Schmidhuber, 1997) but surprisingly worse than the lossless compression algorithm FLAC.

(II) Similar to conclusions within image modeling (Maaløe et al., 2019; Vahdat & Kautz, 2020; Child, 2021), the LVM expressiveness increases with a deeper hierarchy of stochastic latent variables. In direct comparisons, the introduced model outperforms its deterministic counterparts. However, due to computational cost, it remains infeasible to run the model on the same setting as a state-of-the-art WaveNet model.

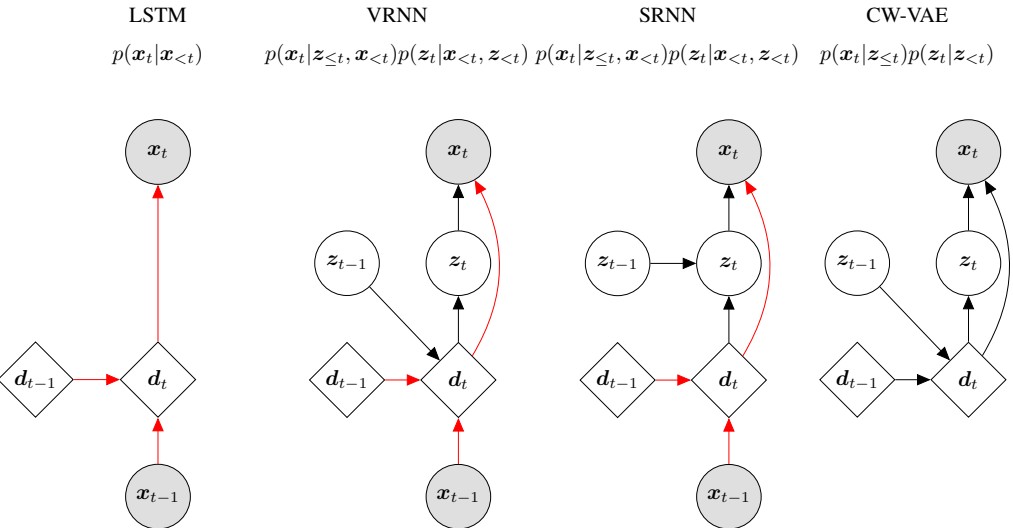

Figure 1: Generative models for a single time step of a deterministic autoregressive LSTM, the VRNN and SRNN, and the CW-VAE with one layer of latent variables. Red arrows indicate purely deterministic paths from the output $x_t$ to previous input $x_{<t}$ without passing a stochastic node. Since the CW-VAE is not autoregressively dependent on the observed variable and inference injects information into only the stochastic variables, information cannot flow through such a fully deterministic path. This is possible in the VRNN and SRNN. We provide graphical illustrations of inference models for VRNN and SRNN in the appendix and for CW-VAE in figure 2.

(III) The introduced model finds strongly clustered speech-related features in its hierarchy of latent variables building a strong case for utilizing such models for other tasks such as semi-supervised learning.

This shows that LVMs without powerful autoregressive decoders for the observed variable have potential as generative speech models when using expressive hierarchies of latent variables.

## 2 LATENT VARIABLE MODELS FOR SPEECH

### 2.1 RELATED WORK

LVMs formulated in context of the VAE framework continue to be of interest due to their ability to learn an approximation to the posterior distribution over the dataset. The posterior is usually on a significantly reduced dimensionality compared to the input data and lies very close to a known prior distribution. Such approximated posterior provides use cases for other tasks beyond generation such as semi-supervised learning (Kingma et al., 2014) and out-of-distribution detection (Havtorn et al., 2021). Furthermore, from image modeling research, we know that powerful LVMs can achieve state-of-the-art performance without costly autoregressive dependencies on the observed variable.

In recent years, there have been several complementary ways of improving the expressiveness of the VAE such as building more expressive priors through methods such as Normalizing Flows (Rezende & Mohamed, 2015) and building a deeper hierarchy of stochastic latent variables such as the Ladder VAE (Sønderby et al., 2016). In this research, we choose to focus on the latter due to the recent breakthroughs resulting in state-of-the-art VAEs without costly autoregressive dependencies on the observed variable (Maaløe et al., 2019; Vahdat & Kautz, 2020; Child, 2021).

To date, there are two widely cited, and to our knowledge state-of-the-art, explicit likelihood generative LVMs for speech:

- The Variational Recurrent Neural Network (VRNN) (Chung et al., 2015).
- The Stochastic Recurrent Neural Network (SRNN) (Fraccaro et al., 2016).

Other recent LVM contributions also achieve impressive results. Among the most noteworthy are the FH-VAE (Hsu et al., 2017), that leverages another stochastic latent variable to capture global latent features in the speech, and the VQ-VAE (Oord et al., 2018b), that introduces a hybrid between an LVM with a quantized latent space and an autoregressive model to generate improved empirical samples. However, the FH-VAE, with its disjoint latent variables, and the VQ-VAE, with its quantized latent space autoregressive prior fitted after training the encoder/decoder, introduce significant changes to the original VAE framework to function. The Stochastic WaveNet (Lai et al., 2018) and STCN (Aksan & Hilliges, 2019) are fully convolutional models that resemble the VRNN. They are however only autoregressive in observed space and utilize a hierarchy of latent variables.

Building on learnings from the LVM improvements in the image domain, we formulate a novel temporal LVM by introducing a hierarchy of stochastic latent variables through the adaptation of a model recently proposed for video prediction:

- The Clockwork Variational Autoencoder (CW-VAE) (Saxena et al., 2021).

## 2.2 TEMPORAL VARIATIONAL AUTOENCODING

The VRNN, SRNN and CW-VAE are all autoencoders and take as input a variable-length sequence $\boldsymbol{x} = (\boldsymbol{x}_1, \boldsymbol{x}_2, \ldots, \boldsymbol{x}_{T_x})$ with $\boldsymbol{x}_t \in \mathcal{X}^{D_x}$. In general, $\boldsymbol{x}$ may refer to the original observed variable or a deterministic and temporally downsampled representation of the observed variable.

First, $\boldsymbol{x}$ is encoded to a temporal stochastic latent representation $\boldsymbol{z} = (\boldsymbol{z}_1, \boldsymbol{z}_2, \ldots, \boldsymbol{z}_{T_z})$ with $\boldsymbol{z}_t \in \mathcal{Z}^{D_z}$ and length $T_z \leq T_x$. This representation is then used to reconstruct the original input $\boldsymbol{x}$. The latent variable is assumed to follow some prior distribution $p(\boldsymbol{z}_t|\cdot)$. The prior distribution may depend on latent and observed variables at previous time steps, $\boldsymbol{z}_{<t}$ and $\boldsymbol{x}_{<t}$, but not $\boldsymbol{x}_t$. Here we have introduced the shorthand notation $\boldsymbol{z}_{<t} = (\boldsymbol{z}_0, \boldsymbol{z}_1, \ldots, \boldsymbol{z}_{t-1})$.

The models are trained to maximize a likelihood objective. The exact likelihood is given by

$$\log p_{\boldsymbol{\theta}}(\boldsymbol{x}) = \log \int p_{\boldsymbol{\theta}}(\boldsymbol{x}, \boldsymbol{z}) \, d\boldsymbol{z} \ , \tag{1}$$

but is intractable to optimize due to the integration over the latent space. Instead, the true posterior is variationally approximated by $q_{\boldsymbol{\phi}}(\boldsymbol{z}|\boldsymbol{x})$ which yields the well-known evidence lower bound (ELBO) on the exact likelihood given by

$$\log p_{\boldsymbol{\theta}}(\boldsymbol{x}) \geq \mathbb{E}_{\boldsymbol{z} \sim q_{\boldsymbol{\phi}}(\boldsymbol{z}|\boldsymbol{x})} \left[ \log p_{\boldsymbol{\theta}}(\boldsymbol{x}, \boldsymbol{z}) - \log q_{\boldsymbol{\phi}}(\boldsymbol{z}|\boldsymbol{x}) \right] = \mathcal{L}(\boldsymbol{\theta}, \boldsymbol{\phi}; \boldsymbol{x}) \ , \tag{2}$$

with respect to $\{\boldsymbol{\theta}, \boldsymbol{\phi}\}$. We omit the $\boldsymbol{\theta}$ and $\boldsymbol{\phi}$ subscripts for the remainder of the paper. A graphical illustration of the models can be seen in figure 1.

## 2.3 VARIATIONAL RECURRENT NEURAL NETWORK (VRNN)

The VRNN (Chung et al., 2015) is essentially a VAE per timestep $t$. Each VAE is conditioned on the hidden state of an RNN $\boldsymbol{d}_{t-1} \in \mathbb{R}^{D_d}$, with state transition $\boldsymbol{d}_t = f([\boldsymbol{x}_{t-1}, \boldsymbol{z}_{t-1}], \boldsymbol{d}_{t-1})$ where $[\cdot, \cdot]$ denotes concatenation. The joint distribution over observed and latent variables factorizes over time and the latent variables are autoregressive in both the observed and latent space:

$$p(\boldsymbol{x}, \boldsymbol{z}) = \prod_t p(\boldsymbol{x}_t|\boldsymbol{z}_{\leq t}, \boldsymbol{x}_{<t}) p(\boldsymbol{z}_t|\boldsymbol{x}_{<t}, \boldsymbol{z}_{<t}) \ . \tag{3}$$

The approximate posterior distribution similarly factorizes over time:

$$q(\boldsymbol{z}|\boldsymbol{x}) = \prod_t q(\boldsymbol{z}_t|\boldsymbol{x}_{\leq t}, \boldsymbol{z}_{<t}) \ . \tag{4}$$

The ELBO then becomes

$$\log p(\boldsymbol{x}) \geq \mathbb{E}_{\boldsymbol{z} \sim q(\boldsymbol{z}|\boldsymbol{x})} \left[ \sum_t \log p(\boldsymbol{x}_t|\boldsymbol{z}_{\leq t}, \boldsymbol{x}_{<t}) - \mathrm{KL}\left( q(\boldsymbol{z}_t|\boldsymbol{x}_{\leq t}, \boldsymbol{z}_{<t}) \, \| \, p(\boldsymbol{z}_t|\boldsymbol{x}_{<t}, \boldsymbol{z}_{<t}) \right) \right] \ . \tag{5}$$

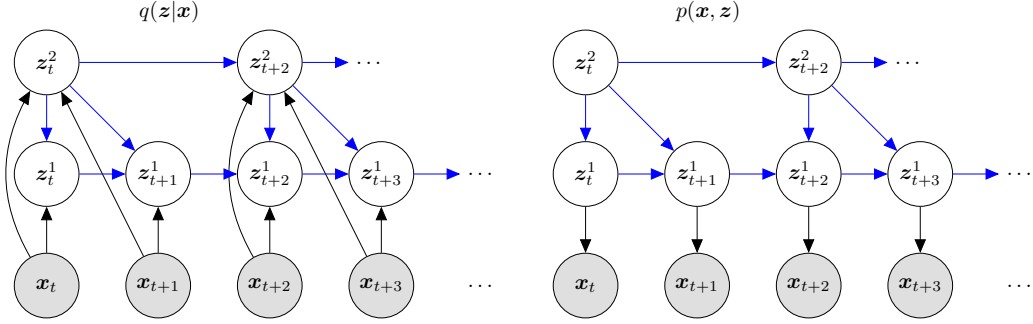

Figure 2: Inference (left) and generative (right) models for the Clockwork VAE with a hierarchy of two latent variables with $s_1 = 1$ and $s_2 = 2$. The models are unrolled over four consecutive time steps but note that the graph continues towards $t = 0$ and $t = T_x$. Blue arrows indicate parameter sharing between the inference and generative models. We omit the deterministic variable for clarity.

The VRNN uses isotropic Gaussian distributions for the prior and posterior. It uses multilayer perceptrons (MLPs), denoted by $\varphi$, to parameterize all distributions.

$$q(\boldsymbol{z}_t|\boldsymbol{x}_{\leq t}, \boldsymbol{z}_{<t}) = \mathcal{N}\left(\boldsymbol{\mu}_{z,t}, \mathrm{diag}(\boldsymbol{\sigma}_{z,t}^2)\right) , \qquad \text{where } [\boldsymbol{\mu}_{z,t}, \boldsymbol{\sigma}_{z,t}^2] = \varphi_{\mathrm{vrnn}}^{\mathrm{enc}}(\boldsymbol{x}_t, \boldsymbol{d}_t) , \qquad (6)$$

$$p(\boldsymbol{z}_t|\boldsymbol{x}_{<t}, \boldsymbol{z}_{<t}) = \mathcal{N}\left(\boldsymbol{\mu}_{0,t}, \mathrm{diag}(\boldsymbol{\sigma}_{0,t}^2)\right) , \qquad \text{where } [\boldsymbol{\mu}_{0,t}, \boldsymbol{\sigma}_{0,t}^2] = \varphi_{\mathrm{vrnn}}^{\mathrm{prior}}(\boldsymbol{d}_t) , \qquad (7)$$

$$p(\boldsymbol{x}_t|\boldsymbol{z}_{\leq t}, \boldsymbol{x}_{<t}) = \mathcal{D}\left(\boldsymbol{\rho}_{x,t}\right) , \qquad \text{where } \boldsymbol{\rho}_{x,t} = \varphi_{\mathrm{vrnn}}^{\mathrm{dec}}(\boldsymbol{z}_t, \boldsymbol{d}_t) . \qquad (8)$$

The recurrent transition function $f$ is parameterized by a Gated Recurrent Unit (GRU) (Cho et al., 2014). At timestep zero, $\boldsymbol{d}_0$ is chosen as the zero vector. $\mathcal{D}(\boldsymbol{\rho}_{x,t})$ denotes any output distribution parameterized by a set of parameters $\boldsymbol{\rho}_{x,t}$.

We note that since the decoder is dependent on $\boldsymbol{d}_t$, the transition function $f$ allows the VRNN to learn to ignore parts of or the entire latent variable and establish a purely deterministic transition from $\boldsymbol{x}_{t-1}$ to $\boldsymbol{d}_t$ similar to a regular GRU (see figure 1) which well-know weakness of VAEs with powerful decoders (Bowman et al., 2016; Sønderby et al., 2016).

### 2.4 STOCHASTIC RECURRENT NEURAL NETWORK (SRNN)

The SRNN (Fraccaro et al., 2016) is similar to the VRNN but differs by separating the stochastic latent variables from the deterministic representations entirely (see figure 1). In generation, this is done by having a GRU run forwards in time over the observed variable to form a deterministic representation $\boldsymbol{d}_t$ from $\boldsymbol{x}_{<t}$. The latent variable is then sampled from the prior $p(\boldsymbol{z}_t|\boldsymbol{x}_{<t}, \boldsymbol{z}_{<t})$ which is conditioned directly on the previous latent variable.

The SRNN also uses a more intricate inference network which essentially learns to solve a smoothing problem rather than a filtering problem by also running backwards in time. Specifically, in the smoothing configuration, the inference model $q(\boldsymbol{z}_t|\boldsymbol{x}_{\leq t}, \boldsymbol{z}_{<t})$ includes an additional deterministic variable computed from $\boldsymbol{d}_t$ and $\boldsymbol{x}_t$ by a GRU running backwards in time i.e. $\boldsymbol{a}_t = f^{\mathrm{rev}}([\boldsymbol{x}_t, \boldsymbol{d}_t], \boldsymbol{a}_{t+1})$. In the filtering configuration, this is replaced with an MLP, $\boldsymbol{a}_t = f^{\mathrm{rev}}([\boldsymbol{x}_t, \boldsymbol{d}_t])$. The encoding distribution $q(\boldsymbol{z}_t|\boldsymbol{x}_{\leq t}, \boldsymbol{z}_{<t})$ is then conditioned on $\boldsymbol{a}_t$, $\varphi_{\mathrm{srnn}}^{\mathrm{enc}}(\boldsymbol{x}_t, \boldsymbol{a}_t)$. In our experiments we run the SRNN in the smoothing configuration.

### 2.5 CLOCKWORK VARIATIONAL AUTOENCODER (CW-VAE)

The CW-VAE (Saxena et al., 2021) is a hierarchical latent variable model recently introduced for video generation. Contrary to the VRNN and SRNN, it is designed to make use of a hierarchy of latent variables and uses top-down inference (Sønderby et al., 2016) which enables learning a covariance between the latent variables in the hierarchy. As illustrated in figure 1, it is autoregressive in the latent space but not in the observed space. Since inference only injects information into the stochastic variables, there is no deterministic path connecting the previously observed variables to the next. This forces the information to flow through the stochastic latent variables (figure 1). Hence, the VRNN and SRNN can collapse to a deterministic model while the CW-VAE cannot.

We denote the latent variable at timestep $t$ and layer $l \in [1, L]$ by $\boldsymbol{z}_t^l$. In CW-VAE each latent layer is updated only every $s_l$ timesteps, where $s_l$ is a layer-dependent integer, or stride, defined in a way such that $s_l$ is largest for larger values of $l$. This imposes the inductive bias that latent variables exist at different temporal resolutions with $\boldsymbol{z}^l$ changing over longer time scales than $\boldsymbol{z}^{l-1}$. In speech, phonetic variation $(10 - 400\,\mathrm{ms})$, morphological and semantic features at the word level and speaker-related variation at the global scale make this a reasonable assumption.

The timesteps at which a layer updates its latent state are given by $\mathcal{T}_l \equiv \{t \in [1, T] \,|\, t \mod s_l = 1\}$. In practice and equivalently, we represent this by having references to the unique states copied over time, $\boldsymbol{z}_t^l \equiv \boldsymbol{z}_{\max_\tau \{\tau \in \mathcal{T}_l | \tau \leq t\}}^l$. The joint distribution factorizes over time and over the latent hierarchy.

$$p(\boldsymbol{x}, \boldsymbol{z}) = \left( \prod_t p(\boldsymbol{x}_t | \boldsymbol{z}_t^1) \right) \left( \prod_{l=1}^{L} \prod_{t \in \mathcal{T}_l} p(\boldsymbol{z}_t^l | \boldsymbol{z}_{t-1}^l, \boldsymbol{z}_t^{l+1}) \right) . \tag{9}$$

The inference model similarly factorizes over time and over the layers of the latent hierarchy with the posterior conditioned on a span of the observed variable $\boldsymbol{x}_{t:t+s_l}$ dependent on the layer stride $s_l$.

$$q(\boldsymbol{z} | \boldsymbol{x}) = \prod_{l=1}^{L} q(\boldsymbol{z}^l | \boldsymbol{x}) = \prod_{l=1}^{L} \prod_{t \in \mathcal{T}_l} q(\boldsymbol{z}_t^l | \boldsymbol{z}_{t-1}^l, \boldsymbol{z}_t^{l+1}, \boldsymbol{x}_{t:t+s_l}) . \tag{10}$$

Similar to the VRNN and SRNN, an encoder, $\varphi_{\mathrm{cwvae}}^{\mathrm{enc}, l}$, is used to parameterize the approximate posterior $q(\boldsymbol{z} | \boldsymbol{x})$ which is taken to be an isotropic Gaussian, as is the prior. This encoder may be the same for all stochastic layers $l$, be layer-specific or use a ladder-network as in the LadderVAE (Sønderby et al., 2016). A graphical representation of the inference and generative models for a two-layered CW-VAE can be seen in figure 2.

## 2.6 MODELING SPEECH WITH CLOCKWORK VAES

In the original paper, $s_l \triangleq k^{l-1}$ for some constant $k$ making it exponentially dependent on the layer index $l$ with $s_1 = 1$. This is feasible in the video domain with typical frame rates of 30 or 60 Hz. In the speech domain however, the observed variable can have sample rates of tens of thousands of frames per second. This results in sequence lengths that are not generally feasible to model with recurrent architectures. For this reason, we chose $s_1 \gg 1$ to achieve an initial temporal downsampling and let $s_l \triangleq c^{l-1} s_1$ for $l > 1$ and some constant $c$.

We design the encoder as a ladder-network similar to Sønderby et al. (2016), as this provides some benefits compared to alternatives such as a standalone encoder per latent variable. Specifically, a ladder-network leverages parameter sharing across the latent hierarchy and importantly processes the full observed sequence only once and shares the resulting representations for all latent variables. This yields a more computationally efficient encoder and a higher activity in latent variables towards the top of the hierarchy. Finally, we parameterize the encoder/decoder networks using 1D convolutions that operate on the raw waveform.

## 2.7 OUTPUT DISTRIBUTION

The choice of output distribution $\mathcal{D}(\boldsymbol{\rho}_{x,t})$ is generally data dependent. At the most fundamental level, we select either continuous and discrete distributions depending on the nature of $\boldsymbol{x}$, the likelihoods of which are not comparable.

In the original VRNN and SRNN, $\mathcal{D}(\boldsymbol{\rho}_{x,t})$ was taken to be the continuous isotropic Gaussian or a mixture of it. This choice generally results in an ill-posed problem with a likelihood that is unbounded from above (Mattei & Frellsen, 2018). As a result, reported likelihoods can be very sensitive to hyperparameter settings and hard to compare. Additionally, in audio modeling, $\boldsymbol{x} \in \{0, 1, \ldots, 2^b - 1\}$ denotes a discrete-valued waveform sampled with some bit-depth $b$. Continuous distributions, and especially mixtures of continuous distributions, are well-known to be able to yield arbitrarily high likelihoods when used to model discrete data (Bishop, 2006). In the VRNN and SRNN, $\boldsymbol{x}$ was scaled to take values between -1 and 1, as is usual, yielding a gap between the discrete values of $\frac{1}{2^b - 1}$. This alleviates the issue with discrete values since $\boldsymbol{x}$ becomes approximately continuous as $b$ becomes large. However, the bit-depth of audio is rarely above 24 meaning that the

gap between unique discrete values remains much larger than e.g. the gap between 32 bit floating point numbers. This ill-posed likelihood has also been used in other works (Hsu et al., 2017; Lai et al., 2018; Aksan & Hilliges, 2019; Zhu et al., 2020).

To correctly model discrete $x$ with a continuous distribution we must dequantize $x$ to be continuous by e.g. adding uniform noise or using a variational approach (Ho et al., 2019). The continuous likelihood obtained via dequantization has been shown to be a lower bound on the likelihood that could have been obtained with a discrete distribution (Theis et al., 2016). Dequantization was common in the image domain (Dinh et al., 2015; Sønderby et al., 2016) until the introduction of the discretized mixture of logistics (DMoL) (Salimans et al., 2017).

The DMoL was introduced for use in autoregressive models (Salimans et al., 2017) but has become the de facto standard output distribution for generative modeling of natural images. Notably, this is also the case for latent variable models that are not autoregressive in the observed variable (Maaløe et al., 2019; Vahdat & Kautz, 2020; Child, 2021). Recently, it was applied to generative speech modeling of raw waveforms (Oord et al., 2018a). As opposed to e.g. a categorical distribution, the DMoL induces ordinality on the observed space such that values that are numerically close are also considered close in the probabilistic sense. This is a sensible inductive bias for images as well as audio where individual samples represent the amplitude of light and pressure, respectively.

## 3 SPEECH MODELING EXPERIMENTS

**Data** We train models on TIMIT (Garofolo, 1993), LibriSpeech (Panayotov et al., 2015) and LibriLight (Kahn et al., 2020). We provide more details on the datasets in the appendix. We represent the audio as $\mu$-law encoded PCM standardized to (discrete) values in $[-1, 1]$ with discretization gap of $\frac{1}{2^{b-1}}$. We use the original bit depth of $16$ bits and sample rate of $16\,000$ Hz. We use this representation both as the input and the reconstruction target.

**Likelihood** We report likelihoods in units of bits per frame (bpf) as this yields a more intuitive, interpretable and comparable version of likelihood than total likelihood in nats. It also has direct connections with information theory and compression (Shannon, 1948; Townsend et al., 2019). In units of bits per frame, lower is better. The obtained likelihoods can be seen in tables 1 and 2. For LVMs, we report the one-sample ELBO. We describe how to convert likelihood to bpf in the appendix.

| $s$ | Model | Configuration | $\mathcal{L}$ [bpf] |
|---|---|---|---|
| 1 | Uniform | Uninformed | 16.00 |
| 1 | DMoL | Optimal | 15.60 |
| - | FLAC | - | **8.582** |
| 1 | WaveNet | $D_c = 96$ | **10.88** |
| 1 | LSTM | $D_d = 256$ | 11.11 |
| 1 | VRNN | $D_z = 256$ | $\leq$11.09 |
| 1 | SRNN | $D_z = 256$ | $\leq$11.19 |
| 64 | WaveNet | $D_c = 96$ | 13.30 |
| 64 | LSTM | $D_d = 256$ | 13.34 |
| 64 | VRNN | $D_z = 256$ | $\leq$12.54 |
| 64 | SRNN | $D_z = 256$ | $\leq$12.42 |
| 64 | CW-VAE | $D_z = 96, L = 1$ | $\leq$12.44 |
| 64 | CW-VAE | $D_z = 96, L = 2$ | $\leq$12.17 |
| 64 | CW-VAE | $D_z = 96, L = 3$ | $\leq$**12.15** |
| 256 | WaveNet | $D_c = 96$ | 14.11 |
| 256 | LSTM | $D_d = 256$ | 14.20 |
| 256 | VRNN | $D_z = 256$ | $\leq$13.27 |
| 256 | SRNN | $D_z = 256$ | $\leq$13.14 |
| 256 | CW-VAE | $D_z = 96, L = 1$ | $\leq$13.11 |
| 256 | CW-VAE | $D_z = 96, L = 2$ | $\leq$12.97 |
| 256 | CW-VAE | $D_z = 96, L = 3$ | $\leq$**12.87** |

Table 1: Model likelihoods $\mathcal{L}$ on TIMIT represented as a 16bit $\mu$-law encoded PCM for different stochastic latent variable models compared to deterministic autoregressive baselines. For the CW-VAE, $s$ refers to $s_1$ and the multi-layered models have $c = 8$. Likelihoods are given in units of bits per frame (bpf).

**Models** Architecture and training details are sketched below, while the full details are in the appendix along with additional results for some alternative model configurations. We select model configurations that can be trained on GPUs with a maximum of 12GB of RAM and train all models until convergence. We use a DMoL with 10 components for the output distribution of all models and model all datasets at their full bit depth of $16$ bits. We supply some additional results with a Gaussian output distribution in the appendix.

| $s$ | Model | Configuration | Likelihood $\mathcal{L}$ [bpf] | | | |
|---|---|---|---|---|---|---|
| | | | dev-clean 10h/100h | dev-other 10h/100h | test-clean 10h/100h | test-other 10h/100h |
| 1 | Uniform | Uninformed | 16.00 | 16.00 | 16.00 | 16.00 |
| 1 | DMoL | Optimal | 15.66 | 15.70 | 15.62 | 15.71 |
| - | FLAC | - | **9.390** | **9.292** | **9.700** | **9.272** |
| 1 | Wavenet | $D_c = 48$ | 11.10/10.92 | 11.02/10.80 | 11.26/11.08 | 11.22/11.00 |
| 1 | Wavenet | $D_c = 96$ | **10.96/10.89** | **10.85/10.76** | **11.12/11.01** | **11.05/10.85** |
| 64 | LSTM | $D_d = 256$ | 13.65/13.49 | 13.62/13.48 | 13.64/13.47 | 13.65/13.49 |
| 64 | CW-VAE | $L = 1$ | $\leq$ 12.32/12.24 | 12.32/12.23 | 12.43/12.33 | 12.43/12.33 |
| 64 | CW-VAE | $L = 2$ | $\leq$ **12.30/12.22** | **12.30/12.21** | **12.40/12.31** | **12.39/12.32** |

Table 2: Model likelihoods $\mathcal{L}$ on LibriSpeech test sets represented as $16$ bit $\mu$-law encoded PCM. For the CW-VAE, $s$ refers to $s_1$ and the two-layered models have $s_2 = 8s_1$. The models are trained on either the $10$ h LibriLight subset or the $100$ h LibriSpeech train-clean-100 subset for comparison. Likelihoods are given in units of bits per frame (bpf).

We supply three elementary baselines that form approximate upper and lower bounds on the likelihood. Specifically, we evaluate an uninformed discrete uniform distribution and a two-component DMoL distribution fitted to the training set to benchmark worst case performance. We also report the compression achieved by the lossless compression algorithm, FLAC (Coalson & Castro Lopo, 2019), which constitutes an approximate lower bound for all considered models.

We configure the WaveNet baseline as in the original paper using ten layers per block and five blocks, $L = 10, B = 5$. We select the number of channels in the convolutions throughout the model, $D_c$, to allow running it on a 12GB GPU. We evaluate WaveNet on single frames $s = 1$ as well as stacks of $s = 64$ and 256 frames.

We also provide an LSTM baseline (Hochreiter & Schmidhuber, 1997) which runs on stacked waveforms with stack size $s$. Hence, every $d_t$ is computed from $x_{t:t+s}$. It uses encoders and decoders similar to the VRNN and SRNN models and differs only in the design of the recurrent cell where the LSTM is fully deterministic. We report on LSTM models with hidden state size $D_d = 256$ here. Other sizes yielded as good or inferior results and can be found in the appendix.

The configuration of the VRNN and SRNN models is with waveform stacks of size $s = 1$, $s = 64$ and 256. The stack size of $s = 1$ is computationally demanding and hence we train this on short randomly sampled segments for each training example and only train them on TIMIT. For both models we set the latent variable equal in size to the hidden units and run them with $D_z = 256$.

The CW-VAE is configured similarly to the VRNN and SRNN models with a convolutional encoder/decoder using strides of $s_1 = 64$ or 256 and using $c = 8$. This yields $s_2 = 512$ or 2048 and $s_3 = 4096$ or 16384, respectively. We run the Clockwork VAE with $L = 1, 2$ and 3 layers of latent variables. The number of convolutional channels is set equal to $D_z$.

### 3.1 RESULTS

For temporal resolutions of $s = 1$, the deterministic autoregressive models yield the best likelihoods with WaveNet achieving $10.88$ bpf on TIMIT as seen in table 1. Somewhat surprisingly, the LSTM baseline almost matches WaveNet with a likelihood of $11.11$ bpf at $s = 1$. Unsurprisingly, due to being autoregressive also in training, the LSTM trains considerably slower than the parallel dilated convolutional architecture of WaveNet. Notably, the VRNN and SRNN models achieve likelihoods close to that of WaveNet and the LSTM at around $11.09$ bpf.

For $s = 64$, the LSTM trains more efficiently but this comes at the cost of higher bpf. Notably, the LSTM at $s = 64$ yields a considerably worse likelihood than do VRNN, SRNN and CW-VAE at the same stride being separated by close to $1$ bit. The CW-VAE outperforms both LSTM, VRNN and SRNN when configured with a hierarchy of latent variables reaching $12.15$ bits with $L = 3$. With a single layer of latent variables, the missing autoregression in the observed space results in

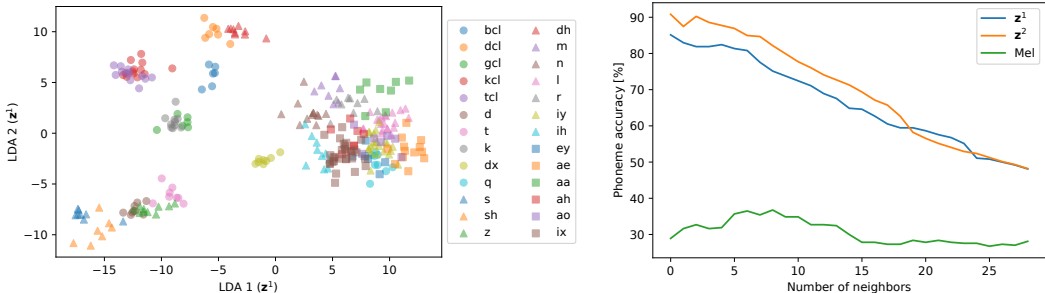

Figure 3: (left) Clustering of phonemes in a two-dimensional Linear Discriminant Analysis (LDA) subspace of a CW-VAE latent space of $z^1$. (right) A leave-one-out classification accuracy for a $k$-nearest-neighbor classifier for different $k$ fitted to a 5D linear subspace of a CW-VAE latent space.

a likelihood of $12.44\,\mathrm{bpf}$ which is inferior to both SRNN and VRNN but notably still better than the LSTM. These observations also carry to $s = 256$, where a multilayered CW-VAE outperforms LSTM, VRNN and SRNN. For strides $s > 1$, previous work has attributed the inferior performance of autoregressive sequence models without latent variables, such as WaveNet and the LSTM, to the ability of LVMs to model intra-step correlations (Lai et al., 2019).

For the CW-VAE, VRNN and SRNN, decreasing the stack size $s$ improves the likelihood, as for deterministic models. This seems to indicate that LVMs may be able to outperform autoregressive models as the stack size approaches $s = 1$. However, due to computational cost of scaling up the CW-VAE for $s = 1$, it was infeasible to perform this experiment at present.

For LibriSpeech (table 2), we see similar results with the CW-VAE improving upon the LSTM with $s = 64$ and the CW-VAE performing better with an additional layer of latent variables. WaveNet performs similar on LibriSpeech compared to TIMIT This shows that state-of-the-art likelihoods can be achieved for speech generation with autoregression purely in the latent space as long as the latent space is expressive enough. This is similar to the findings on state-of-the-art LVMs for image modeling (Maaløe et al., 2019; Vahdat & Kautz, 2020; Child, 2021).

Finally, a connection can be made between the likelihoods achieved by the considered models and the compression rates of lossless audio compression algorithms. Whereas lossy compression algorithms such as MP3 exploit the dynamic range of human hearing to achieve 70-95% reduction in bit rate (Brandenburg et al., 1998), state-of-the-art lossless compression algorithms such as FLAC achieve 50-70% (Coalson & Castro Lopo, 2019) independently of audio content. The LibriSpeech dataset achieves an overall compression to 56.2% of the original bit rate. Although both the deterministic autoregressive models and the LVMs are lossy, the objective they are trained towards minimize the amount of incurred loss which arguably makes them comparable to lossless compression algorithms. The best likelihoods achieved by the models considered above roughly correspond to a 30% reduction in bit rate which seems to indicate that there are still solid gains in likelihood to be made in speech modeling.

## 3.2 EVALUATION OF LATENT REPRESENTATIONS

A potential benefit of LVMs over purely autoregressive models is that they learn a distilled representation of the data in the form of latent variables. These should capture the high-level information in the data, and may serve as building blocks for downstream tasks. Here we evaluate the quality of this learned latent space.

**Phonemes** Phonemes are a fundamental unit of speech that relate to how certain parts of a word are pronounced. The TIMIT dataset is annotated with temporally aligned phoneme classes which allows us to analyze how they are organized in the CW-VAE latent space. Phonemes have durations in the milliseconds which for the TIMIT dataset ranges between $10$ and $400\,\mathrm{ms}$ (see figure 11 in appendix for a full overview). Phoneme recognition is a form of automatic speech recognition and is closely related to the speech-to-text task (Hsu et al., 2017).

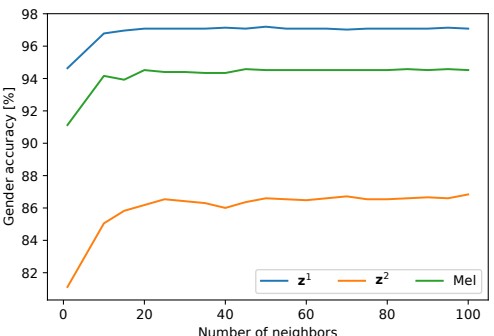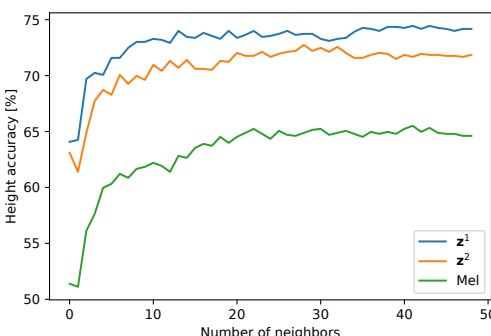

Figure 4: Leave-one-out $k$-nearest-neighbor accuracy with different $k$ for (left) the speaker's gender and (right) the height of male speakers (female speakers yield a similar result).

We perform the evaluation with a two-layered CW-VAE with $s_1 = 64$ and $s_2 = 512$. These strides correspond to temporal resolutions of $4\,\mathrm{ms}$ and $32\,\mathrm{ms}$ both of which are on the order of phonemes. We therefore expect both latent variables to capture information related to phonemes. We infer the latent representations of all utterances of a single speaker from the TIMIT test set. We take 100 Monte Carlo samples to estimate the mean representation per time step. We then compute the average latent representation over the duration of each phoneme. This approximately marginalizes out other sources of variation within the duration of the phoneme. We obtain a low-dimensional linear subspace of the latent space using a linear discriminant analysis (LDA). By considering a single speaker, we exclude inter-speaker variation in phoneme pronunciation.

In figure 3 (left) we visualize the linear subspace of $z^1$ and the resulting phoneme clusters along with the average accuracy of a leave-one-out $k$-nearest-neighbor (KNN) classifier on the single left-out latent representation reduced with a 5-dimensional LDA. We compare to the time-averaged frequency-bin representation of a Mel-spectrogram, also LDA reduced, with hop length set to 64, equal to $s_1$ for the CW-VAE, window size 256 and 80 Mel bins. This is a very common preprocessing step in speech applications. We note that most phonemes are separable in the subspace and that related phonemes such as "s" and "sh" are close[1]. We show that both latent spaces yield significantly better KNN accuracies than the Mel features indicating their usefulness for downstream tasks.

**Exploration of latent variable hierarchy** In figure 4 we similarly compute a leave-one-out KNN classification on the time-averaged latent representations and Mel-features for the gender and height of the speaker. We divide the height into three classes: below $175\,\mathrm{cm}$, above $185\,\mathrm{cm}$ and in-between. Compared to phonemes, these are global attributes that can affect the qualities of speech. Here we again see improved performance from using the learned latent space over Mel-features. Notably, $z^2$ is outperformed by the Mel-features for gender identification which may indicate that this latent variable learns to ignore this attribute compared to $z^1$.

## 4 CONCLUSION/DISCUSSION

In this paper we have presented benchmarks for speech generation using stochastic latent variable models in comparison to deterministic autoregressive models. We have adapted the state-of-the-art video generation model, Clockwork VAE, to the speech domain, similar to how WaveNet adapted the PixelCNN. The Clockwork VAE with a hierarchy of stochastic latent variables outperformed other latent variable models that are autoregressive on both the observed and latent variables. This is in itself impressive since the Clockwork VAE is autoregressive only in the latent space. Finally, the Clockwork VAE outperformed a high-performing deterministic autoregressive model showing comparable results to WaveNet. However, we are still to see a latent variable model outperform WaveNet in a direct comparison. The Clockwork VAE current remains too computationally expensive for this to happen. This research serves as a first step in that direction.

---

[1]A full description of the phonemes and phone codes used for the TIMIT dataset can be found at `https://catalog.ldc.upenn.edu/docs/LDC93S1/PHONCODE.TXT`

REPRODUCIBILITY STATEMENT

The source code used for the work presented in this paper will be made available before the conference. This code provides all details, practical and otherwise, needed to reproduce the results in this paper including data preprocessing, model training, model likelihood and latent space evaluation. The source code also includes scripts for downloading and preparing the LibriSpeech, LibriLight and TIMIT datasets. The LibriSpeech and LibriLight datasets are open source and can be downloaded with the preparation scripts. They are also available at `https://www.openslr.org/12` and `https://github.com/facebookresearch/libri-light`, respectively. The TIMIT dataset is commercial and must be purchased and downloaded from `https://catalog.ldc.upenn.edu/LDC93S1` before running the preparation script.

The stochastic latent variable models considered in this work do not provide an exact likelihood estimate nor an exact latent space representation. For the likelihood, they provide a stochastic lower bound and some variation in the reproduced likelihoods as well as latent representations must be expected between otherwise completely identical forward passes. This variance is fairly small in practice when averaging over large datasets such as those considered in this work. We seed our experiments to reduce the randomness to a minimum, but parts of the algorithms underlying the CUDA framework are stochastic for efficiency. To retain computational feasibility, we do not run experiments with a deterministic CUDA backend.

ETHICS STATEMENT

The work presented here fundamentally deals with automated perception of speech and generation of speech. These applications of machine learning potentially raise a number of ethical concerns. For instance, the these models might see possibly adverse use in automated surveillance and generation of deep fakes. To counter some of these effects, this work has focused on openness by using publicly available datasets for model development and benchmarking. Additionally, the work will open source the source code used to create these results. Ensuring the net positive effect of the development of these technologies is and must continue to be an ongoing effort.

We do not associate any significant ethical concerns with the datasets used in this work. However, one might note that the TIMIT dataset has somewhat skewed distributions in terms of gender and race diversity. Specifically, the male to female ratio is about two to one while the vast majority of speakers are Caucasian. Such statistics might have an effect of some ethical concern on downstream applications derived from such a dataset as also highlighted in recent research (Koenecke et al., 2020). In LibriSpeech, there is an approximately equal number of female and male speakers while the diversity in race is unknown to the authors.

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

# A   DATASETS

**TIMIT**   TIMIT (Garofolo, 1993) is a speech dataset which contains $16\,\text{kHz}$ recordings of 630 speakers of eight major dialects of American English, each reading ten phonetically rich sentences. It amounts to 6300 total recordings splits approximately in 3.94 hours of audio for training and 1.43 hours of audio for testing. No speakers or sentences in the test set are in the training set. TIMIT includes temporally aligned annotations of phonemes and words as well as speaker metadata such as gender, height, age, race, education level and dialect region (Garofolo, 1993).

**LibriSpeech and LibriLight**   The LibriSpeech dataset (Panayotov et al., 2015) consists of readings of public domain audio books amounting to approximately $1000\,\text{h}$ of audio. The data is derived from the LibriVox project. LibriLight (Kahn et al., 2020) is a subset of LibriSpeech created as an automatic speech transcription (ASR) benchmark with limited or no supervision. We specifically train on the $100\,\text{h}$ train-clean-100 subset of LibriSpeech and the $10\,\text{h}$ subset of LibriLight.

Both datasets store the audio as $16\,\text{bit}$ pulse code modulation (PCM).

# B   MODEL ARCHITECTURES

**VRNN**   We implement the VRNN as described in the original work (Chung et al., 2015) but replace the Gaussian output distribution with the DMoL.

**SRNN**   We implement the VRNN as described in the original work (Fraccaro et al., 2016) but replace the Gaussian output distribution with the DMoL.

**CW-VAE**   We implement the CW-VAE based on the original work (Saxena et al., 2021) but with some modifications also briefly described in section 2.6. We replace the encoder/decoder model architectures of the original work with architectures designed for waveform modelling. Specifically, the encoder and decoder are based on the Conv-TasNet (Luo & Mesgarani, 2019) and uses similar residual block structure. However, contrary to the Conv-TasNet, we require downsampling factors larger than two. In order to achieve this we

**LSTM**   The LSTM baseline uses an MLP encoder to embed the waveform subsegment $\boldsymbol{x}_{t:t+s-1}$ to a feature vector before feeding it to the LSTM cell. The encoder is similar to the parameterization of $\phi_{\text{vrnn}}^{\text{enc}}$ for the VRNN described above. The LSTM cell produces the hidden state $\boldsymbol{d}_t$ from $\boldsymbol{x}_{t:t+s-1}$ and passes it to a decoder. Like the encoder, the decoder is parameterized like $\phi_{\text{vrnn}}^{\text{dec}}$ of the VRNN. It outputs the waveform predictions $\boldsymbol{x}_{t+s:t+2s-1}$ from the hidden state $\boldsymbol{d}_t$.

**WaveNet**   We implement WaveNet as described in the original work (Oord et al., 2016a) but use a discretized mixture of logistics as the output distribution as also done in other work (Oord et al., 2018a). Our WaveNet is not conditioned on any signal other than the raw waveform. The model applies the causal convolution directly to the raw waveform frames (i.e. one input channel). An alternative option that we did not examine is to replace the initial convolution with an embedding lookup with a learnable vector for each waveform frame value.

During training, a single WaveNet output is independent from inputs outside of the receptive field. For this reason, we can generally train WaveNet on random subsequences of training examples that are shorter than the full examples. This reduces memory requirements but does not bias the gradient. The subsequences are chosen to be of length 16000 for models with $s = 1$ which is larger than the receptive field of 5117 and corresponds to one second of audio in TIMIT and LibriSpeech. For models with $s = 64$ and $s = 256$ we train on the full example lengths since the receptive field is effectively $s$ times larger.

# C   TRAINING DETAILS

We implement all models and training scripts in PyTorch 1.9 (Paszke et al., 2017). For both datasets we use the Adam optimizer (Kingma & Ba, 2015) with default parameters as given in PyTorch. We

use learning rate $3e-4$ and no learning rate schedule. We use PyTorch automatic mixed preceision (AMP) for WaveNet to yield significant reductions in memory consumption. We did not observe any significant difference in final model performance compared to full ($32\,\mathrm{bit}$) precision when trained with a small number of channels.

We split TIMIT as done in other work (Chung et al., 2015; Fraccaro et al., 2016; Aksan & Hilliges, 2019). We train and test on the full examples, padding batches with zeros when examples are not of equal length. We sample batches such that they consist of examples that are approximately the same length to minimize the amount of computation wasted on padding.

We train on the 10h and 100h hour subsets of Librispeech (Panayotov et al., 2015). The 10h subset is as defined in the LibriLight dataset (Kahn et al., 2020) (a subset of Librispeech). The 100h subset corresponds to the Librispeech train-clean-100 split. In all cases we evaluate on all the test splits dev-clean, dev-other, test-clean, test-other.

Since the average sequence length in Librispeech is about 4 times longer than that of TIMIT, we do not train on the full sequences. Instead, in each epoch we randomly sample a segment of length $L_s$ frames from each training example. For WaveNet we set $L_s = 16000$ which, as required, is larger than the receptive field and train with batch size 2. For CW-VAE we set $L_s = 128000$ as the model is autoregressive in latent space and as such can benefit from longer sequence lengths and train with batch size 4. This length approximately corresponds to the length of the longest examples in TIMIT.

In testing, we evaluate on the full sequences. For LibriSpeech we need to split the test examples into segments, again due to memory constraints. Hence, we do several forward passes per test example, carrying along the internal state for models that are autoregressive in training (LSTM, VRNN, SRNN, CW-VAE) and define segments to overlap according to model architecture.

## D    CONVERTING THE LIKELIHOOD TO UNITS OF BITS PER FRAME

Here we briefly describe how to compute a likelihood in units of bits per frame (bpf). In the main text, we use $\log$ to mean $\log_e$, but here we will be explicit. In general, conversion from nats to bits (i.e., from $\log_e$ to $\log_2$) is achieved by $\log_2(x) = \log_e(x)/\log_2(e)$. Remember that $\log_2 p(\boldsymbol{x})$ factorizes as $\sum_t \log_2 p(\boldsymbol{x}_t)$. In contrast to computing bits per dimension in the image domain, it is important to remember that each example $\boldsymbol{x}^i$ must be weighted differently according the sequence length of each specific example. Thus, we compute the likelihood in bits per frame over the entire dataset as:

$$\frac{\sum_i \sum_t \log_2 p(\boldsymbol{x}_t^i)}{\sum_i T_i} \tag{11}$$

where $i$ denotes the example index, $T_i$ is the length of that example and $t$ is the time index.

## E    ADDITIONAL LATENT SPACE CLUSTERING

We provide some additional latent space clustering of speaker gender in figure 5 and of speaker height in figure 6.

## F    ADDITIONAL LIKELIHOOD RESULTS

**TIMIT, $\mu$-law, DMoL**    We provide additional results on TIMIT with audio represented as $\mu$-law encoded PCM in table 3.

**TIMIT, linear, Gaussian**    We also provide some results on TIMIT with the audio instead represented as linear PCM (linearly encoded) and using Gaussian output distributions as has been done previously in the literature (Chung et al., 2015; Fraccaro et al., 2016; Lai et al., 2018; Aksan & Hilliges, 2019). We provide the results in table 4 and include likelihoods reported in the literature for reference. For our models, we use the same architectures as before but replace the discretized mixture of logistics with either a Gaussian distribution or a mixture of Gaussian distributions.

We constrain the variance of the Gaussians used with our models to be at least $\sigma^2_{\min} = 0.01^2$ in order to avoid the variance going to zero, the likelihood going to infinity and optimization becoming unstable.

From table 4 we note that the performance of the CW-VAE with Gaussian output distribution when modeling linear PCM (i.e. not $\mu$-law encoded) does not compare as favorably to the other baselines as it did with the discretized mixture of logistics distribution. We hypothesize that this has to do with using a Gaussian output distribution in latent variable models which, as has been reported elsewhere (Mattei & Frellsen, 2018), leads to a likelihood function that is unbounded above and can grow arbitrarily high. We discuss this phenomenon in further detail in section G.

We specifically hypothesize that models that are autoregressive in the observed variable (VRNN, SRNN, Stochastic WaveNet, STCN) are well-equipped to utilize local smoothness to put very high density on the correct next value and that this in turn leads to a high degree of exploitation of the unboundedness of the likelihood. Not being autoregressive in the observed variable, the CW-VAE cannot exploit this local smoothness in the same way. Instead, the reconstruction is conditioned on a stochastic latent variable, $p(\boldsymbol{x}_t|\boldsymbol{z}^1_t)$, which introduces uncertainty and likely larger reconstruction variances.

## G  ADDITIONAL DISCUSSION ON GAUSSIAN LIKELIHOODS IN LVMs

As noted in section F, we constrain the variance of the output distribution of our models to be $\sigma^2_{\min} = 0.01^2$ for the additional results on TIMIT with Gaussian outputs. This limits the maximum value attainable by the prediction/reconstruction density of a single waveform frame $x_t$.

Specifically, we can see that since

$$\log p(x_t|\cdot) = \log \mathcal{N}\left(x_t; \mu_t, \max\left\{\sigma^2_{\min}, \sigma^2_t\right\}\right) \ , \tag{12}$$

the best prediction/reconstruction density is achieved when $\sigma^2 \leq \sigma^2_{\min}$ and $\mu = x_t$. Here $\cdot$ indicates any variables we might condition on such as the previous input frame $x_{t-1}$ or some latent variables. We can evaluate this best case scenario for $\sigma^2_{\min} = 0.01^2$,

$$\begin{aligned}\log \mathcal{N}\left(x_t; x_t, \sigma^2_{\min}\right) &= -\frac{1}{2}\log 2\pi - \frac{1}{2}\log \sigma^2_{\min} - \frac{1}{2\sigma^2_{\min}}(x_t - x_t) \\ &= -\frac{1}{2}\log 2\pi - \frac{1}{2}\log 0.01^2 \\ &= 3.686 \ . \end{aligned} \tag{13}$$

Hence, with perfect prediction/reconstruction and the minimal variance ($0.01^2$), a waveform frame contributes to the likelihood with $3.686\,\text{nats}$. With an average test set example length of $49\,367.3\,\text{frames}$ frames this leads to a best-case likelihood of $181967$. We provide a list of maximally attainable Gaussian likelihoods on TIMIT for different minimal variances in table 5. One can note that the maximal likelihood at $\sigma^2_{\min} = 0.1^2$ is lower than the likelihoods achieved by some models in table 4. This indicates that the models learn to use very small variances in order to increase the likelihood.

## H  ADDITIONAL DISCUSSION ON THE CHOICE OF OUTPUT DISTRIBUTION

The DMoL uses a discretization of the continuous logistic distribution to define a mixture model over a discrete random variable. This allows it to parameterize multimodal distributions which can express ambiguity about the value of $\boldsymbol{x}_t$. The model can learn to maximize likelihood by assigning a bit of probability mass to multiple potential values of $\boldsymbol{x}_t$.

While this is well-suited for autoregressive modeling, for which the distribution was developed, the potential multimodality poses a challenge for non-autoregressive latent variable models which independently sample multiple neighboring observations at the output. In fact, if multiple neighboring outputs defined by the subsequence $\boldsymbol{x}_{t_1:t_2}$ have multimodal $p(\boldsymbol{x}_t|\cdot)$, we risk sampling a subsequence where each neighboring value expresses different potential realities, independently.

| $s$ | **Model** | **Configuration** | $\mathcal{L}$ **[bpf]** |
|---|---|---|---|
| 1 | Wavenet | $D_C = 16$ | 11.27 |
| 1 | Wavenet | $D_C = 24$ | 11.14 |
| 1 | Wavenet | $D_C = 32$ | 11.03 |
| 1 | Wavenet | $D_C = 96$ | 10.88 |
| 1 | Wavenet | $D_C = 128$ | 10.98 |
| 1 | Wavenet | $D_C = 160$ | 10.91 |
| 1 | LSTM | $D_d = 128, L = 1$ | 11.40 |
| 1 | LSTM | $D_d = 256, L = 1$ | 11.11 |
| 1 | VRNN | $D_z = 256$ | $\leq 11.09$ |
| 1 | SRNN | $D_z = 256$ | $\leq 11.19$ |
| 4 | LSTM | $D_d = 256, L = 1$ | 11.65 |
| 16 | LSTM | $D_d = 256, L = 1$ | 12.54 |
| 16 | LSTM | $D_d = 256, L = 2$ | 12.54 |
| 16 | LSTM | $D_d = 256, L = 3$ | 12.44 |
| 64 | WaveNet | $D_c = 96$ | 13.30 |
| 64 | LSTM | $D_d = 96, L = 1$ | 13.49 |
| 64 | LSTM | $D_d = 96, L = 2$ | 13.46 |
| 64 | LSTM | $D_d = 96, L = 3$ | 13.40 |
| 64 | LSTM | $D_d = 256, L = 1$ | 13.27 |
| 64 | LSTM | $D_d = 256, L = 2$ | 13.29 |
| 64 | LSTM | $D_d = 256, L = 3$ | 13.31 |
| 64 | LSTM | $D_d = 512, L = 1$ | 13.37 |
| 64 | LSTM | $D_d = 512, L = 2$ | 13.37 |
| 64 | LSTM | $D_d = 512, L = 3$ | 13.41 |
| 64 | VRNN | $D_z = 96$ | $\leq 12.93$ |
| 64 | VRNN | $D_z = 256$ | $\leq 12.54$ |
| 64 | SRNN | $D_z = 96$ | $\leq 12.87$ |
| 64 | SRNN | $D_z = 256$ | $\leq 12.42$ |
| 64 | CW-VAE | $D_z = 96, L = 1$ | $\leq 12.44$ |
| 64 | CW-VAE | $D_z = 96, L = 2$ | $\leq 12.17$ |
| 64 | CW-VAE | $D_z = 96, L = 3$ | $\leq 12.15$ |
| 64 | CW-VAE | $D_z = 256, L = 2$ | $\leq 12.10$ |
| 256 | WaveNet | $D_c = 96$ | 14.11 |
| 256 | LSTM | $D_d = 256, L = 1$ | 14.20 |
| 256 | LSTM | $D_d = 256, L = 2$ | 14.17 |
| 256 | LSTM | $D_d = 256, L = 3$ | 14.26 |
| 256 | VRNN | $D_z = 96$ | $\leq 13.51$ |
| 256 | VRNN | $D_z = 256$ | $\leq 13.27$ |
| 256 | SRNN | $D_z = 96$ | $\leq 13.28$ |
| 256 | SRNN | $D_z = 256$ | $\leq 13.14$ |
| 256 | CW-VAE | $D_z = 96, L = 1$ | $\leq 13.11$ |
| 256 | CW-VAE | $D_z = 96, L = 2$ | $\leq 12.97$ |
| 256 | CW-VAE | $D_z = 96, L = 3$ | $\leq 12.87$ |

Table 3: Model likelihoods on TIMIT represented as a 16bit $\mu$-law encoded PCM, obtained by different latent variable models and compared to autoregressive baselines all using a discretized mixture of logistics with 10 components as output distribution. Likelihoods are given in units of bits per frame (bpf) and obtained by normalizing the total likelihood of each sequence with the individual sequence length and then averaging over the dataset.

Interestingly, most work on latent variable models with non-autoregressive output distributions seem to ignore this fact and simply employ the mixture distribution with 10 mixture components (Maaløe et al., 2019; Vahdat & Kautz, 2020; Child, 2021). However, given the empirically good results

| $s$ | Model | Configuration | $\mathcal{L}$ [nats] |
|-----|-------|---------------|--------|
| 1 | WaveNet | Normal | 74962 |
| 200 | WaveNet (Aksan & Hilliges, 2019) | GMM | 30188 |
| 200 | WaveNet (Aksan & Hilliges, 2019) | Normal | -7443 |
| 200 | Stochastic WaveNet* (Lai et al., 2018) | Normal | $\geq$72463 |
| 200 | VRNN (Chung et al., 2015) | Normal | $\approx$28982 |
| 200 | SRNN (Fraccaro et al., 2016) | Normal | $\geq$60550 |
| 200 | STCN (Aksan & Hilliges, 2019) | GMM | $\geq$69195 |
| 200 | STCN (Aksan & Hilliges, 2019) | Normal | $\geq$64913 |
| 200 | STCN-dense (Aksan & Hilliges, 2019) | GMM | $\geq$71386 |
| 200 | STCN-dense (Aksan & Hilliges, 2019) | Normal | $\geq$70294 |
| 200 | STCN-dense-large (Aksan & Hilliges, 2019) | GMM | $\geq$77438 |
| 200 | CW-VAE* | $L = 1, D_z = 96$, Normal | $\geq$41629 |

Table 4: Model likelihoods on TIMIT represented as globally normalized 16bit linear PCM. Likelihoods are given in units of nats and obtained by summing over the likelihood all examples in the dataset and dividing by the sum of all their sequence lengths. In the table, Normal refers to using a Gaussian likelihood and GMM refers to using a Gaussian Mixture Model likelihood with 20 components. Models with asterisks * are our implementations while remaining results are as reported in the referenced work.

| $\sigma^2_{\min}$ | max $\mathcal{L}$ |
|-------------------|-------------------|
| $1^2$ | -45367 |
| $0.5^2$ | -11146 |
| $0.1^2$ | 68307 |
| $0.05^2$ | 102525 |
| $0.01^2$ | 181979 |
| $0.005^2$ | 216198 |
| $0.001^2$ | 295651 |

Table 5: The highest possible Gaussian log-likelihoods (max $\mathcal{L}$) attainable on TIMIT as computed by equation 12 with different values of the minimum variance $\sigma^2_{\min}$.

of latent variable models for image generation, this seems to have posed only a minor problem in practice. We speculate that this is due to the high degree of similarity between neighbouring pixels in images. I.e. if the neighboring pixels are nuances of red, then, in all likelihood, so is the central pixel.

In the audio domain, however, neighbouring waveform frames can take wildly different values, especially at low sample rates. Furthermore, waveforms exhibit a natural symmetry between positive and negative amplitudes. Hence, it seems plausible that multimodality may pose a larger problem in non-autoregressive speech generation by causing locally incoherent samples than it seems to do in image modelling.

Finally, one can note that for continuous variables, the change of variables formula can in many cases convert the likelihood of a random variable following one distribution into the likelihood of another variable following a different distribution. However, the change of variables formula does not apply to conversions where one or both random variables are discrete.

## I    ADDITIONAL GRAPHICAL MODELS

In figure 7 we show the unrolled graphical model of a three-layered CW-VAE with $k_1 = 1$ and $c = 2$ yielding $k_2 = 2$ and $k_3 = 4$. We show both the generative and inference models and highlight in blue the parameter sharing between the two models due to top-down inference.

In figure 8 we show the graphical model of the recurrent cell of the CW-VAE for a single time step. As noted in (Saxena et al., 2021), this cell is very similar to the one of the Recurrent State Space Model (RSSM) (Hafner et al., 2019).

In figure 9 we illustrate the unrolled graphical models of the inference and generative models of the VRNN (Chung et al., 2015). We include the deterministic variable $d_t$ in order to illustrate the difference to other latent variable models.

Likewise, in figure 10 we illustrate the unrolled graphical models the SRNN (Fraccaro et al., 2016).

## J DISTRIBUTION OF PHONEME DURATION IN TIMIT

In figure 11 we plot a boxplots of the duration of each phoneme in the TIMIT dataset. We do this globally as well as for a single speaker to show that phoneme duration can vary between individual speakers.

## K CW-VAE SAMPLES FROM THE PRIOR

For the two-layered CW-VAE trained on TIMIT, we provide samples from the prior at the following URL: `https://doi.org/10.5281/zenodo.5704513`

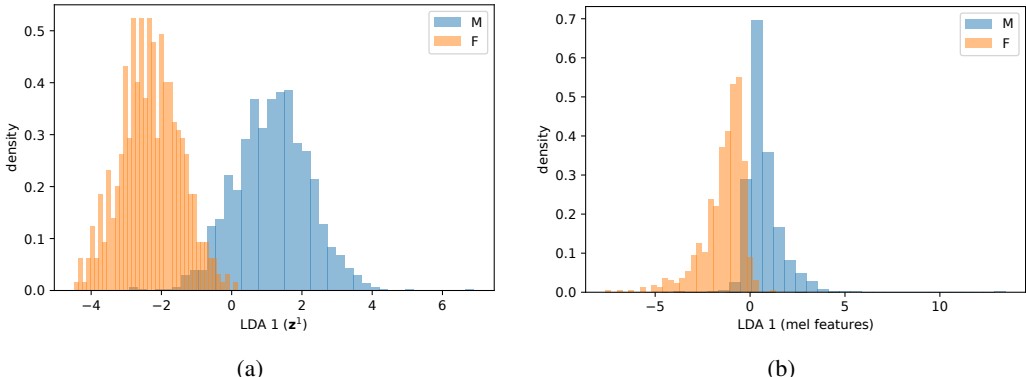

(a)                                             (b)

Figure 5: Clustering of speaker gender in an one-dimensional linear subspace defined by a linear discriminant analysis of the CW-VAE latent space and of a time-averaged mel spectrogram. The total overlap is slightly smaller in the subspace of the CW-VAE latent space and the separation between the distribution peaks is larger.

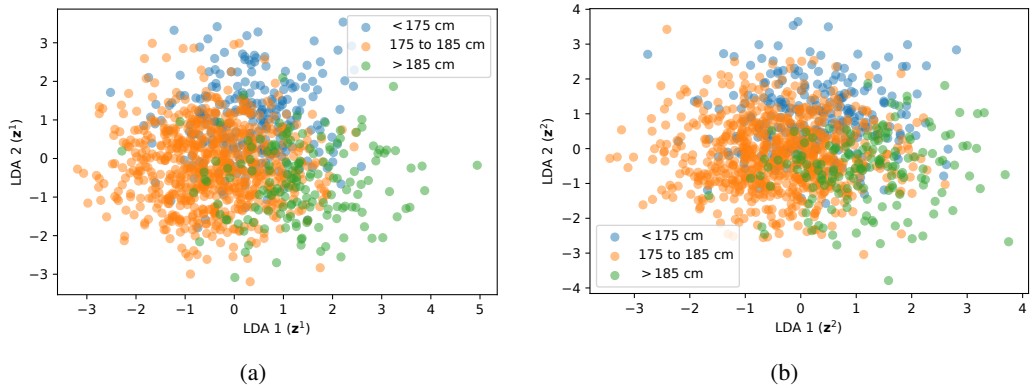

(a)                                             (b)

Figure 6: Clustering of speaker height in an two-dimensional linear subspace defined by a linear discriminant analysis of the CW-VAE latent space.

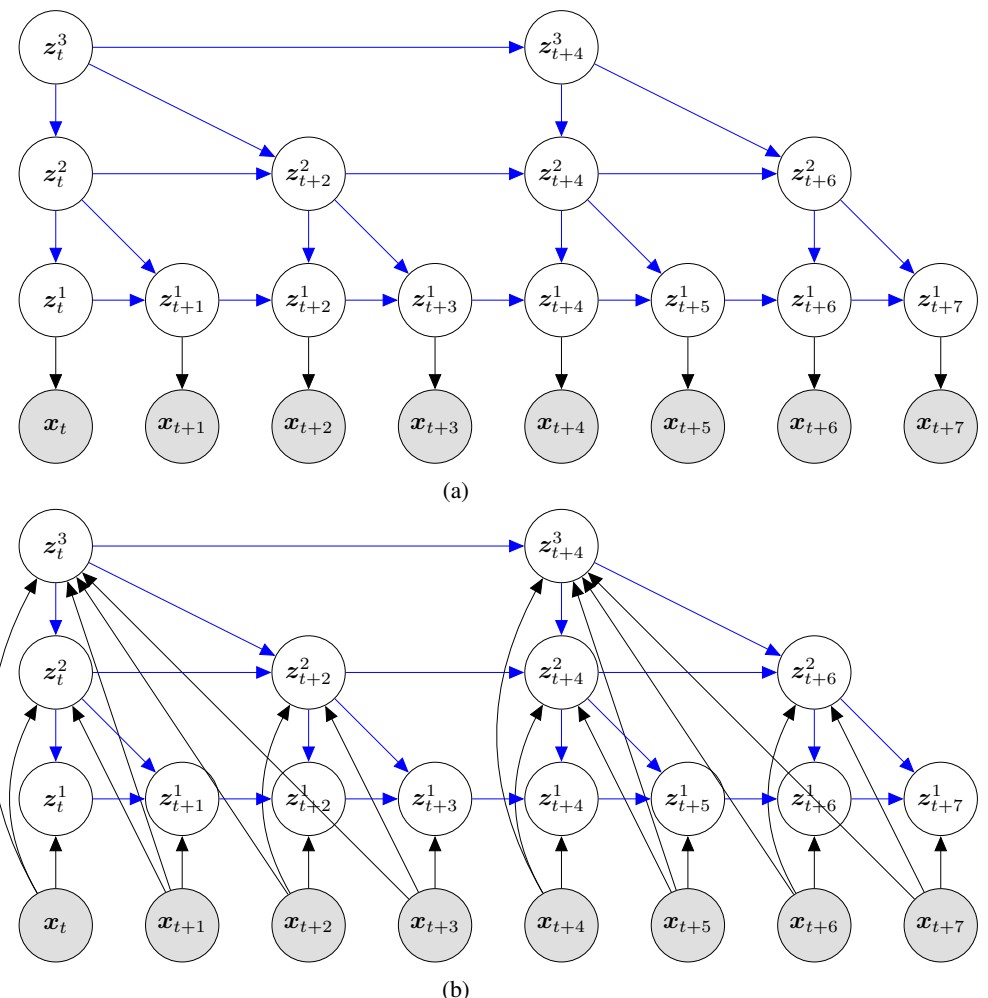

Figure 7: CW-VAE generative model $p(\boldsymbol{x}, \boldsymbol{z})$ in (a) and inference model $q(\boldsymbol{z}|\boldsymbol{x})$ in (b) for a three-layered model with $k_1 = 1$ and $c = 2$ giving $k_2 = 2$ and $k_3 = 4$ unrolled over eight steps in the observed variable. Blue arrows are (mostly) shared between the inference and generative models. See figure 8 for a detailed graphical model expanding on the latent nodes $\boldsymbol{z}_t^l$ and parameter sharing.

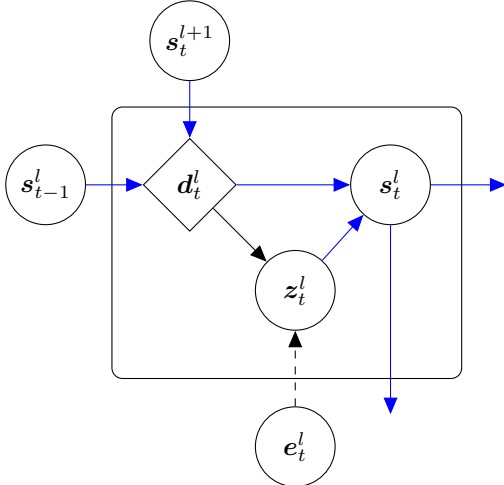

Figure 8: CW-VAE cell state $s_t^l$ update. All blue arrows are shared between generation and inference. The dashed arrow is used only during inference. The solid arrow has unique transformations during inference and generation.

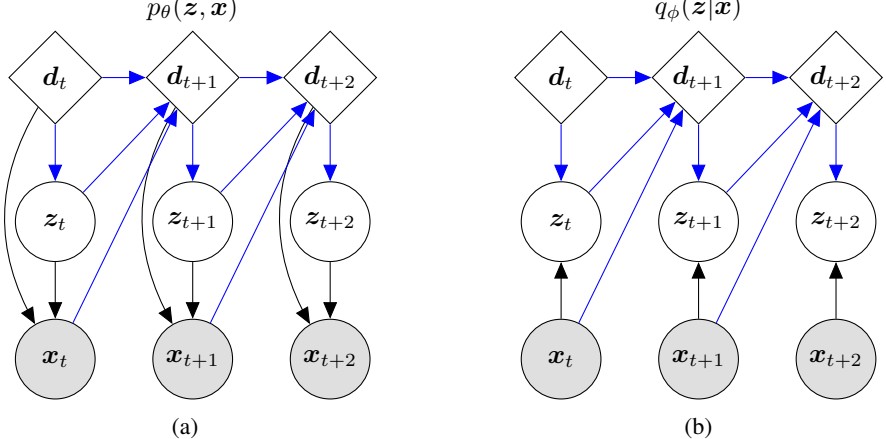

Figure 9: VRNN (Chung et al., 2015) generative model $p(\boldsymbol{x}, \boldsymbol{z})$ in (a) and inference model $q(\boldsymbol{z}|\boldsymbol{x})$ in (b) unrolled over three steps in the observed variable. Blue arrows are shared between the inference and generative models.

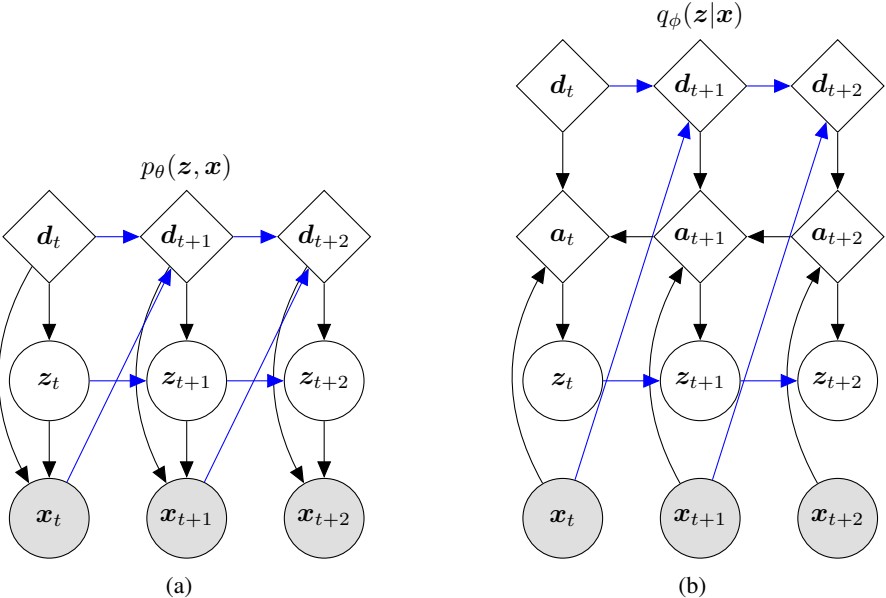

Figure 10: SRNN (Fraccaro et al., 2016) generative model $p(\boldsymbol{x}, \boldsymbol{z})$ in (a) and inference model $q(\boldsymbol{z}|\boldsymbol{x})$ in (b) unrolled over three steps in the observed variable. Blue arrows are shared between the inference and generative models.

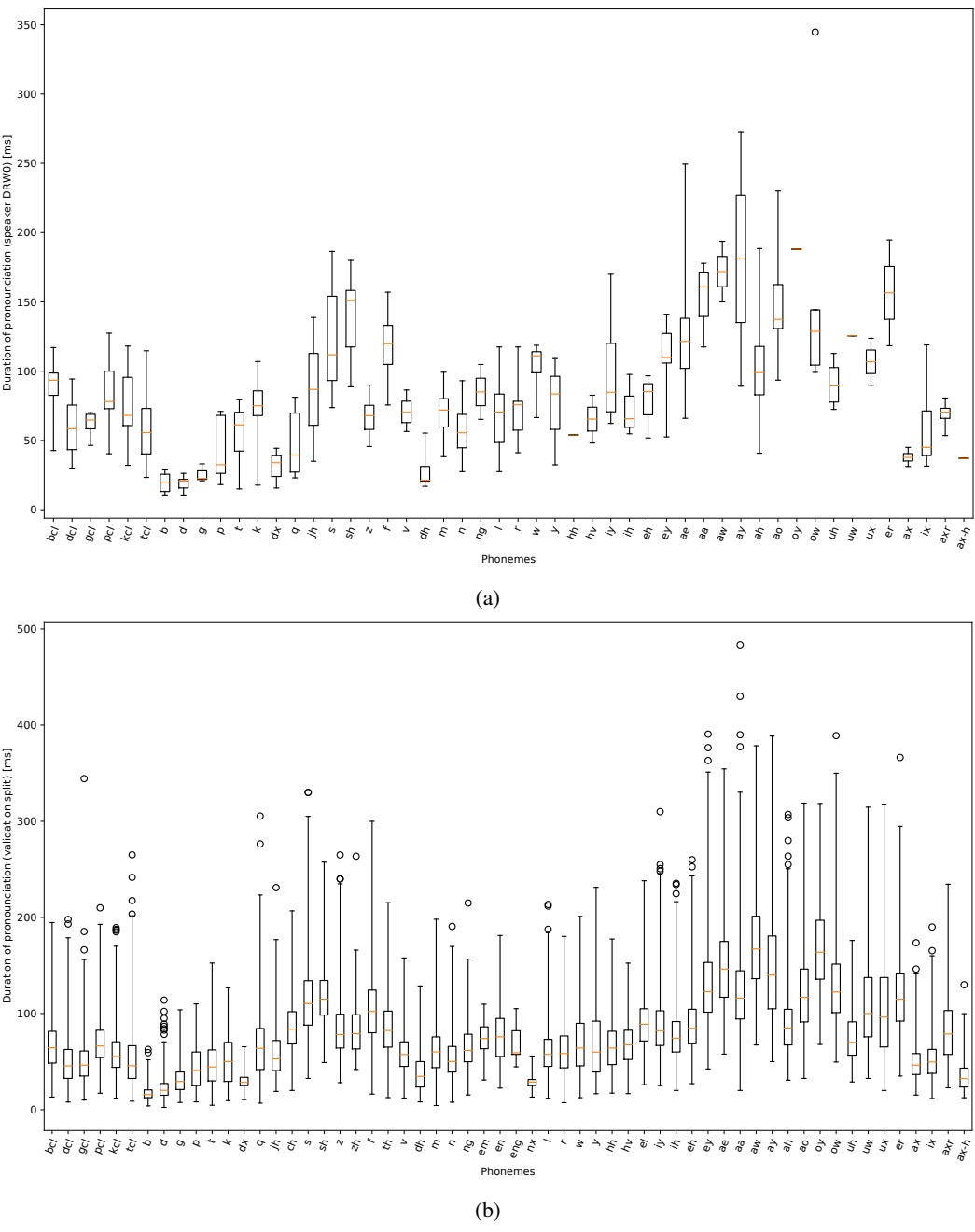

(a)

(b)

Figure 11: Boxplots of the duration of the pronunciation of phonemes in TIMIT for a specific speaker DRW0 in (a) and globally in (b). Not all phonemes are pronounced by speaker DRW0 over the course of their 10 test set sentences and hence they are missing from the x-axis compared to the global durations.

