# OpenReview forum: "Towards Generative Latent Variable Models for Speech"
_ICLR.cc/2022/Conference — ICLR 2022 Submitted_

### Official Review · Reviewer_q4Q4 · 2021-11-02

**Correctness:** 4
**Technical Novelty And Significance:** 2
**Empirical Novelty And Significance:** 3
**Recommendation:** 6
**Confidence:** 3

**Main Review:**

Strengths:  This paper addresses a series of issues that need to be resolved to apply an LV model like clockwork VAE to data like speech.  I find that sequence of steps to be instructive, and the overall target to be one worthy of exploration.  I think that collectively all the engineering described here is a significant amount of work and I appreciate it all being in one place.

Weaknesses: This paper seems quite detached from the community that would find it most interesting.  Speech generation is something that has been studied for a very long time, so I would have expected to at least get some sense of how this approach would compare (on various aspects) with modern practice.  Of course, WaveNet is one model that can serve as a well-recognized benchmark, but there is a lot more to compare with here. I am also uncomfortable with the use of bps as a performance measure of the generative power of these models.  I would have liked to hear some examples and understand how these models differ in their outputs.

**Summary Of The Paper:**

This paper presents an exploration of the use of latent variable models as generative models of speech.  Noting that such models work well in the image space, but not so much in the speech space, the authors move on to adapt the Clockwork VAE (a video LVM) as a speech model.  In the process the authors present a series of useful technical solutions to various issues that arise in this domain transition.  This, and other generative models of speech are later compared in the experiments section.  The results show that this approach is potentially viable.  The performance of the proposed speech LVM is good, albeit it comes with increased computational complexity (hopefully something to solve in the future).  In addition, it is shown that the resulting latent representation is correlated with phonetic structure, which is a pleasant bonus that other speech generative models (e.g. WaveNet) lack.

**Summary Of The Review:**

This paper provides an insightful exploration of how one can use an LVM as a speech generative model.  Although not completely achieved here, I feel that this paper shows some intriguing progress towards that goal, and towards speech models with semantically meaningful latent states.  I think many researchers in this area will find interest in all the engineering that was put to work in this paper, which might also be useful outside of this particular problem.  On the downside, this paper doesn't feel like it addresses any deep scientific questions (although it touches on some near the end), and it mostly reads like a todo list to get clockwork VAE to work with speech.

---

> ### Author Response · Authors · 2021-11-16
> **Response to Reviewer q4Q4**
>
> **Review 3.1**: “Speech generation is something that has been studied for a very long time, so I would have expected to at least get some sense of how this approach would compare (on various aspects) with modern practice. Of course, WaveNet is one model that can serve as a well-recognized benchmark, but there is a lot more to compare with here.” … “On the downside, this paper doesn't feel like it addresses any deep scientific questions (although it touches on some near the end), and it mostly reads like a todo list to get clockwork VAE to work with speech.”
>
> **Response 3.1**: Thank you for this. We have tried to focus on only two widely used deterministic models, the LSTM and WaveNet models, to limit the experimental setup and solely focus on models directly comparable to the LVMs. While we do acknowledge that there are many other deterministic models, we found these two candidates to be good representatives as a first benchmark against which to compare the latent variable models. If you feel that we are missing crucial models, we are happy to include them?
>
> The contribution provides a basis for showing where we are wrt. learning latent variable models on speech. Given the new results, as described in the overall comments, we are equally as far, if not slightly better, as the LSTM and WaveNet. This provides a basis for the “deep scientific questions” to be answered.
>
> The deep scientific question we try to tackle is: “How do we define a latent variable model, that similar to the findings from natural images, can prove state-of-the-art results in generation of speech”. Since the CW-VAE results are proving better than the deterministic and stochastic counterparts, we do believe that we are getting one step closer to answering this question.
>
> **Review 3.2**: I am also uncomfortable with the use of bps as a performance measure of the generative power of these models. I would have liked to hear some examples and understand how these models differ in their outputs.
>
> **Response 3.2**: We completely agree that the likelihood is not a good metric for measuring generative performance as has also been demonstrated in the literature (see e.g. [Theis et al, 2016](https://arxiv.org/pdf/1511.01844.pdf)). Neither do we claim it to be. We provide some samples from the prior of a learned two-layered Clockwork VAE at this anonymized URL: [https://doi.org/10.5281/zenodo.5704513](https://doi.org/10.5281/zenodo.5704513).

---

### Official Review · Reviewer_5Ydq · 2021-11-02

**Correctness:** 4
**Technical Novelty And Significance:** 2
**Empirical Novelty And Significance:** 2
**Recommendation:** 5
**Confidence:** 4

**Main Review:**

The paper is well motivated. I do think benchmarking different latent models is worth doing, and reporting the compression rate is the right metric.

The experiments themselves are fine, but the evaluation metric is a little confusing. All models, except vanilla LSTM and WaveNet, have hidden variables to marginalize, so I'm not entirely sure how the likelihoods are computed. Marginalization is difficult as the paper argued. It's unclear whether, for example, the numbers in Table 1 are simply the values of the variational lower bound, or if any approximation is done to marginalize the hidden variables.

The paper also attempts to answer why one model can be better than the others. The paper looks into phonemes and speaker genders, but the message is not clear.

The presentation is fine. The majority of the paper is spent reviewing the models. I have mixed feelings visualizing the models the way it is done in Figure 1. Figure 2 is a much better representation, laying out the conditional assumptions for both the encoders and decoders. I do understand it would take up a lot of space, but it might be worth putting a figure in the appendix. It's also worth talking about the independence assumptions and where uncertainties are baked in.

**Summary Of The Paper:**

This paper proposes to put various models under the same experimental setting and compare their rate at compressing speech. The models of choice are vanila LSTMs, variational RNNs, stochastic RNNs, Clockwork VAEs, and WaveNets. The results are also compared against regular compression algorithms, for example, FLAC.

**Summary Of The Review:**

The paper is well motivated. The experimental design is fine, but it's unclear how the evaluation metric, the likelihood, is computed without marginalizing the hidden vectors. The presentation is fine.

---

> ### Author Response · Authors · 2021-11-16
> **Response to Reviewer 5Ydq**
>
> **Review 2.1**: “The experiments themselves are fine, but the evaluation metric is a little confusing. All models, except vanilla LSTM and WaveNet, have hidden variables to marginalize, so I'm not entirely sure how the likelihoods are computed. Marginalization is difficult as the paper argued. It's unclear whether, for example, the numbers in Table 1 are simply the values of the variational lower bound, or if any approximation is done to marginalize the hidden variables.”
>
> **Response 2.1**: Indeed you are correct that we did not make this entirely clear. We do not employ any attempts of marginalizing the latent variables besides the usual evidence lower bound with a single Monte Carlo sample. This approach is common within the stochastic latent variable model literature when dealing with sequence data (VRNN, SRNN etc.).
>
> **Review 2.2**: “The paper also attempts to answer why one model can be better than the others. The paper looks into phonemes and speaker genders, but the message is not clear.”
>
> **Response 2.2**: We acknowledge that the message carried by our results on clustering of phonemes, speaker gender and speaker height might not be communicated clearly enough. What we wished to communicate was that the unsupervised discovery of acoustic units related to speech such as phonemes, and the captured correlation of the latent representations with speaker gender and height, shows that the model learns rich representations and exploits the latent variable to encode this information. In latent variable models, these representations are well-defined while in autoregressive models such as WaveNet, such representations are less clearly defined. As a result, autoregressive models are not immediately applicable for down-stream tasks or semi-supervised learning. We will revise this section for the final manuscript. Thank you for this.
>
> **Review 2.3**: “I have mixed feelings visualizing the models the way it is done in Figure 1. Figure 2 is a much better representation, laying out the conditional assumptions for both the encoders and decoders. I do understand it would take up a lot of space, but it might be worth putting a figure in the appendix. It's also worth talking about the independence assumptions and where uncertainties are baked in.”
>
> **Response 2.3**: We agree that graphical illustrations of inference models are of equal importance to the generative models. We were however not able to include an exhaustive illustration including both inference and generative models for all considered model variants in a single figure, or standalone in the main text. We will accommodate this to the best of our abilities by including graphical illustrations of the VRNN and SRNN inference models in the appendix, similar to that included for the CW-VAE in the main text.

---

### Official Review · Reviewer_3vU6 · 2021-11-03

**Correctness:** 2
**Technical Novelty And Significance:** 2
**Empirical Novelty And Significance:** 2
**Recommendation:** 3
**Confidence:** 4

**Main Review:**

Strength:
1. The proposed method is clear. The experiment results support that the proposed method is better than the VRNN and SRNN methods.

Weakness:
1. The novelty is limited. The main ideas about clock-wise RNN network and CWVAE ideas have been proposed.
2. The hierarchy latent variable idea has been proposed in Stochastic WaveNet (https://arxiv.org/pdf/1806.06116.pdf) and STCN (https://arxiv.org/pdf/1902.06568.pdf). This paper didn’t compare the proposed method with these two methods.
3. In the paper, https://arxiv.org/pdf/1902.01388.pdf, the authors point out the evaluation problem of the stochastic sequence neural network. They found the stochastic sequence neural network has an unfair advantage over the deterministic model when s!=1. And with some tricks, the deterministic model can catch up the stochastic model's performance. But the authors didn’t discuss and address this issue in this paper.

**Summary Of The Paper:**

This paper presents a variant of stochastic sequence neural network, the family of VRNN and SRNN. This paper adopts the CW-VAE framework and completes the optimization process under the stochastic sequence neural network framework. The authors test it on the speech domain. The experiments show that it outperforms VRNN and SRNN in the benchmark datasets.

**Summary Of The Review:**

The proposed idea is reasonable but didn't conduct a comprehensive study comparing related works.

---

> ### Author Response · Authors · 2021-11-16
> **Response to Reviewer 3vU6 (continued)**
>
> **Review 1.3**: “In the paper, https://arxiv.org/pdf/1902.01388.pdf, the authors point out the evaluation problem of the stochastic sequence neural network. They found the stochastic sequence neural network has an unfair advantage over the deterministic model when s!=1. And with some tricks, the deterministic model can catch up the stochastic model's performance. But the authors didn’t discuss and address this issue in this paper.”
>
> **Response 1.3**: Thanks for sharing this contribution which we admittedly did not know about beforehand (will be included in the final manuscript). The authors of this paper claim that probabilistic latent variable models applied to sequences have an unfair advantage over similar deterministic models when the observed sequence is modelled in chunks of $s>1$ input frames. Specifically, the authors claim that the unfair advantage stems from probabilistic latent variable models conditioning on a latent variable which allows them to model "intra-step" correlations between the $s$ observed values even though the output distribution is fully factorized over these values. Deterministic models without latent variables do not have this ability.
>
> We do however not agree with the conclusion of this paper that latent variables in general only add to the likelihood of sequence data through this specific mechanism and we ultimately see the inability of autoregressive models to model these “intra-step” correlations as a limitation specific to such models. As we also state in our paper, a more fair comparison takes place at $s=1$ where only a single input frame is modelled making discussions about “intra-step” correlations irrelevant. We have run SRNN and VRNN in this setting (with short input sequence lengths during training to make it feasible) and find that both VRNN and SRNN yield likelihoods comparable to our WaveNet and LSTM baselines. Specifically, VRNN achieves 11.22 and SRNN 11.20 bits per frame which closely matches the performance of the LSTM and WaveNet models.

---

> ### Author Response · Authors · 2021-11-16
> **Response to Reviewer 3vU6**
>
> **Review 1.1**: “The novelty is limited. The main ideas about clock-wise RNN network and CWVAE ideas have been proposed.”
>
> **Response 1.1**: We do agree that these contributions have already been introduced. However, we do also think that a contribution showing that a model introduced on data domain X with some alterations can be successfully applied on data domain Y is novel. Therefore, we did not find it necessary to introduce this as a new model-name, but rather a simple adaptation. See our overall response above.
>
> **Review 1.2**: “The hierarchy latent variable idea has been proposed in Stochastic WaveNet (https://arxiv.org/pdf/1806.06116.pdf) and STCN (https://arxiv.org/pdf/1902.06568.pdf). This paper didn’t compare the proposed method with these two methods.”
>
> **Response 1.2**: Thank you for highlighting this. We did actually analyse these two contributions in-depth and should have included it. Previously, we trained a Wavenet model using the same output distribution as they used and compared it against their tables. The authors of Stochastic Wavenet claim to outperform a vanilla WaveNet but we could not make their benchmark of the essential WaveNet baseline correspond to our findings. The authors report to achieve a likelihood on TIMIT of 26074 with a diagonal covariance Gaussian output distribution and normalized waveform frames as inputs and targets (Table 1). In the same setting, the authors of STCN report likelihoods of WaveNet of -7433 with a diagonal covariance Gaussian and 30188 with a Gaussian mixture distribution with 20 components (Table 4, appendix). A WaveNet modified to be more comparable with STCN (WaveNet-dense) achieves -8579 with a Gaussian and 30636 with a Gaussian mixture. In stark contrast, our implementation of WaveNet is able to achieve a likelihood of 74103 on TIMIT which not only outperforms both respective reimplementations of WaveNet but also Stochastic WaveNet itself and the regular sized versions of STCN (not "large"). Therefore, we didn’t find it necessary to include these papers as comparisons. We should have included a discussion of these papers in our work. This we will do. We will also include a comparison of the Gaussian STCN and Stochastic WaveNet results on TIMIT with our implementation of WaveNet with a Gaussian output distribution in our appendix. Finally, we will include the Clockwork VAE in this comparison.
>
> In terms of methodology, we note that although the Stochastic WaveNet and the STCN define hierarchies of latent variables, all their latent variables operate at the same temporal scale. This stands in contrast with our adaptation of the Clockwork VAE which importantly runs latents at lower sampling rates towards the top of the hierarchy. Another important difference to the previous work in STCN and Stochastic WaveNet is that the Clockwork VAE is autoregressive in latent space and not in observed space. Both STCN and Stochastic WaveNet are autoregressive in observed space and not in latent space. The observation model and structural model are important design choices in latent variable models and hence our adaptation of the Clockwork VAE to speech is a novel contribution in this respect. We also note that both STCN and Stochastic WaveNet are defined with a Gaussian output distribution over audio frames which, as we argue in our paper (see section 2.7), is less well-motivated than using a discrete distribution and subject to certain pathologies, among which is a theoretically unbounded likelihood.

---

### Author Response · Authors · 2021-11-16
**Overall response to all reviewers**

We thank all reviewers for taking the time to provide us with the thorough and constructive feedback that we received. We will first address some important overall themes in the reviews and then provide a response addressing each reviewer in turn. The response to each reviewer is found under the respective review. We hope to hear back from all reviewers before the end of the discussion period.

**Overall response to all reviewers**

- **Novelty of contribution**: This paper serves as the first contribution that provides a hierarchical latent variable model with latent variables operating over different emporal scales of the audio input; an idea that recently has resulted in major performance gains for natural image modeling see e.g. [BIVA](https://arxiv.org/pdf/1902.02102.pdf), [NVAE](https://arxiv.org/pdf/2007.03898.pdf) and [VD-VAE](https://arxiv.org/pdf/2011.10650.pdf). The novelty of presenting a model adapted from a different data domain (in this example video modeling) corresponds to many previous papers that were presumed novel, e.g. WaveNet was simply the adaptation of the PixelCNN/PixelCNN++ model from images to audio. Furthermore, we see it as invaluable to the speech-modeling research community to get proper benchmarks, since previous contributions cannot be used as a reliable basis for comparison (see answers below).
- **New results**: Since submitting the paper, we have modelled the VRNN and SRNN for the $s=1$ case to enable direct comparison with Wavenet (and the LSTM). Here we found that the SRNN scored 11.20 and the VRNN scored 11.22. This shows that current latent variable models can in fact achieve results matching the much used Wavenet architecture. To our knowledge, this is the first time that a contribution has shown that LVMs can in fact match deterministic frameworks for speech in this setting. See the revised Table 1 below (will be updated in the final manuscript):


| $s$  | Model   | Configuration | L [bpf]   |
| ---- | ------- | ------------- | --------- |
| 1    | Uniform | Uninformed    | 16.00     |
| 1    | DMoL    | Optimal       | 15.60     |
| -    | FLAC    |               | **8.582** |
|      |         |               |           |
| 1    | LSTM    |               | 11.11     |
| 1    | WaveNet | $D_C=16$      | 11.27     |
| 1    | WaveNet | $D_C=32$      | 11.03     |
| 1    | WaveNet | $D_C=96$      | **10.88** |
| 1    | WaveNet | $D_C=160$     | 10.91     |
| 1    | VRNN    |               | 11.22     |
| 1    | SRNN    |               | 11.20     |
|      |         |               |           |
| 64   | LSTM    |               | 13.34     |
| 64   | VRNN    |               | 12.54     |
| 64   | SRNN    |               | 12.42     |
| 64   | CW-VAE  | $L=1$         | 12.60     |
| 64   | CW-VAE  | $L=2$         | **12.25** |
|      |         |               |           |
| 256  | LSTM    |               | 14.20     |
| 256  | VRNN    |               | 13.27     |
| 256  | SRNN    |               | 13.14     |
| 256  | CW-VAE  | $L=1$         | 13.11     |
| 256  | CW-VAE  | $L=2$         | **12.97** |

---

### Author Response · Authors · 2021-11-22
**Comment with upload of revised paper**

With this comment we supply a revised version of our paper to reflect the discussion in the comments below. Besides the changes listed in the comments, we have made the following additions to the experimental work:
- Improved previously reported VRNN and SRNN baselines for $s=1$ on TIMIT in table 1.
- Added WaveNet benchmarks for $s=64$ and $s=256$ on TIMIT in table 1.
- Added three-layered Clockwork VAE on TIMIT to table 1.

Additionally, we have made the following minor revisions:
- Changed naming of “bits per sample” to “bits per frame” to avoid confusion with bits per second when bits per sample was abbreviated to bps.
- Reduced the number of WaveNet entries in table 1 to just show the best model. Moved the other entries to the appendix.
- Revised discussion on output distribution and issues related to continuous distributions.
- Revised some formulations for clarity.
- Fixed some typos.
- Updated appendix with additional results, discussion and description of models and training.
- Added ethics statement.

For the camera-ready version we will additionally benchmark the STCN model on TIMIT using a discretized mixture of logistics. This is in order to create a baseline of this model with a discrete output distribution that is more well-behaved than the continuous Gaussian. The discretized mixture of logistics output distribution was not considered in the original work.

---

### Decision · Program_Chairs · 2022-01-20

**Decision:**

Reject

**Comment:**

This paper presents the application of the hierarchical latent variable model, CW-VAE which is originally developed in the vision community, to the speech domain with meaningful modifications, and provide empirical analysis of the likelihood as well as discussions on the likelihood metrics. The reviewers tend to agree that it is a promising direction to study hierarchically structured LVMs for speech, and the introduction/adaptation of CW-VAE is useful. There were some discussion on the suitability of the likelihood evaluation, and it appears a fair comparison with wavenet shall take place at s=1 (single sample), a resolution level the proposed method does not yet scale up to. On the other hand, an important potential use case of the model is representation learning for speech, as it is a common belief that at suitable resolution the features shall discover units like phoneme. But I find the current evaluation of latent representations by LDA and KNN to be somewhat limited, and in fact there is no comparison with suitable baselines in Sec 3.2 in terms of feature quality. A task closer to modern speech recognition (e.g., with end-to-end models) would be preferred.